# Verifiable Delay Function and Its Blockchain-Related Application: A Survey

**DOI:** 10.3390/s22197524

**Published:** 2022-10-04

**Authors:** Qiang Wu, Liang Xi, Shiren Wang, Shan Ji, Shenqing Wang, Yongjun Ren

**Affiliations:** 1Engineering Research Center of Digital Forensics, Ministry of Education, School of Computer Science, Nanjing University of Information Science & Technology, Nanjing 210044, China; 2Beijing Institute of Computer Technology & Application, Beijing 100082, China; 3College of Computer Science & Technology, Nanjing University of Aeronautics & Astronautics, Nanjing 211106, China

**Keywords:** verifiable delay function, blockchain, algebraic assumption, structural assumption

## Abstract

The concept of verifiable delay functions has received attention from researchers since it was first proposed in 2018. The applications of verifiable delay are also widespread in blockchain research, such as: computational timestamping, public random beacons, resource-efficient blockchains, and proofs of data replication. This paper introduces the concept of verifiable delay functions and systematically summarizes the types of verifiable delay functions. Firstly, the description and characteristics of verifiable delay functions are given, and weak verifiable delay functions, incremental verifiable delay functions, decodable verifiable delay functions, and trapdoor verifiable delay functions are introduced respectively. The construction of verifiable delay functions generally relies on two security assumptions: algebraic assumption or structural assumption. Then, the security assumptions of two different verifiable delay functions are described based on cryptography theory. Secondly, a post-quantum verifiable delay function based on super-singular isogeny is introduced. Finally, the paper summarizes the blockchain-related applications of verifiable delay functions.

## 1. Introduction

The concept of a verifiable delay function was first proposed in 2018 by Boneh et al. [1], who proposed several candidate structures for constructing verifiable delay functions and it is an important tool to add time delay in decentralized applications [2,3,4,5]. To be exact, the verifiable delay function is a function f:X→Y that takes a prescribed wall-clock time to compute, even on a parallel processor, ond outputs a unique result that can effectively output the verification. In short, even if it is evaluated on a large number of parallel processors and still requires evaluation of *f* in a specified number of sequential steps. Most importantly, given an input *x* and an output *y*, anyone must quickly verify the output y=f(x). That is to say, for all x∈X and y∈Y, this function f:X→Y satisfies the following requirements:(1)$$x\to x^{2}\to x^{2^{2}}\to x^{2^{3}}\to \ldots \to x^{2^{T}}\;\mathrm{mod}\;N=y.$$

The verifiable delay function is a cryptographic function that requires to be computed in *T* sequential steps and produces a unique, efficiently and publicly verified output [6]. Because the verifiable delay function satisfies the characteristic of sequentiality, the iterated value does depend on the order of the iterated elements. Choose the tuple (N,x,T) as the puzzle, and the verifiable delay function is defined as
(2)$$e:=2^{T}\;\mathrm{mod}\;\varphi \left(N\right),\;y:=x^{e}\;\mathrm{mod}\;N.$$
where N=p·q is an RSA modulus [7], $x\in Z_{N}^{*}$ is a random seed, T∈N is time parameter and knows the group order φ(N)=(p−1)·(q−1).

Although verifiable delay functions have been roughly described in the review of verifiable delay functions published by Boneh et al. [8], the summary is not comprehensive with the emergence of more candidate structures of verifiable delay functions. In addition, the application of different kinds of verifiable delay functions in the blockchain is not explained in detail. Therefore, this paper makes a more comprehensive and detailed summary.

The verifiable delay function has several important characteristics, such as being *T*-sequential, uniqueness and effective verifiability, as shown in Table 1.

The remainder of this paper is organized as follows. Section 2 introduces the descriptions of verifiable delay functions. Section 3 describes verifiable delay functions based on various algebraic assumptions. In Section 4, verifiable delay functions based on various structural assumptions are introduced in detail. Section 5 the postquantum-secure verifiable delay function. Section 6 describes applications of verifiable delay functions combined with blockchain. Section 7 gives a summary.

## 2. Descriptions of Verifiable Delay Functions

The concept of the verifiable delay function first proposed by Boneh and Fisch. The verifiable delay function requires a specified number of sequential steps to evaluate and will produce a function with a unique output that can be validated effectively and publicly. Next, a triple of algorithm (*Setup*, *Eval*, *Verify*) of verifiable delay functions are described as follows [9]. The algorithm flow is shown in Figure 1 and the different types of verifiable delay functions are described in Table 2.

Setup(λ,T)→pp=(ek,vk) is a randomized algorithm that takes a delay parameter *T* and a security parameter λ as input and outputs public parameters pp composed of the evaluation key ek and the verification key vk. Because Setup algorithm is limited by security parameter λ, the running time cannot be too long. In addition, Setup algorithm usually needs a secret random as a parameter to ensure meaningful security, so it is difficult to avoid that the scheme needs a trusted setup to select the random.

Eval(ek,x)→(y,π) is a slow cryptographic algorithm that takes the evaluation key ek and a random seed x∈X as input and outputs a y∈Y together with a possibly empty proof π. To ensure sequentiality, Eval must run in time *T* with no more than a polynomial logarithm of *T* parallel processors.

Verify(vk,x,y,π)→{accept,reject} is a deterministic cryptographic algorithm, in which the algorithm inputs verification key vk, random seed *x*, outputs *y* and proof π. If f(x)=y, output accept; Otherwise output reject. Verify is much faster than Eval and it must run in total time polynomial in log(T) and λ.

### 2.1. Weak Verifiable Delay Functions

**Definition** **1.**
*(Weak verifiable delay functions.) The system V = (Setup, Eval, Verify) is a weak verifiable delay function if the verifiable delay function allows Eval to achieve O(T) parallelism. $\left(T^{2},o\left(T\right)\right)$-sequentiality can only be meaningful for a weak verifiable delay function if Eval is allowed strictly less that T−o(T) on fewer than $T^{2}$ parallel processors, otherwise the honest computation of Eval would require more parallelism than even the adversary is allowed.*


The weak verifiable delay function can be constructed based on the existence of degree *T* injective rational maps [10] that cannot be inverted faster than computing polynomial greatest common denominators of degree *T* polynomials.

Injective rational maps. Define the reverse problem of an injective rational map *F* = ($f_{1},...,f_{m}$) on algebraic sets $Y\subseteq F_{q}^{n}$ to $X\subseteq F_{q}^{m}$, where each $f_{i}:F_{q}^{n}\to F_{q}$ is a rational function in $F_{q}$( $X_{1},...,X_{n}$), for i={1,...,m}. An algebraic set Y is the set of vanish points of some set of polynomial *S*.

Boneh et. al. abstract weak verifiable delay functions from an injective rational map. First, let *F*: $F_{q}^{n}\to F_{q}$ be a rational function that is an injective map from Y to X:=F(Y). At the same time, X is required to be efficiently sampleable and *F* can be evaluated efficiently for all $y^{\prime }\in Y$. If you need to use the injective rational map function *F* in the verifiable delay function, you must guarantee $|X|>\lambda T^{3}$ to prevent brute force attacks, where a delay parameter *T* and a security parameter λ as input to the Setup algorithm.

Verifiable delay functions construct a weak verifiable delay function by function family $F:={\left(q,F,X,Y\right)}_{\lambda ,T}$ with a security parameter λ and a delay parameter *T* as input parameters.

Setup(λ,T)→pp=((q,F),(q,F)) is a randomized algorithm that takes a delay parameter *T* and a security parameter λ as input and choose a (q,F,X,Y)∈F, then outputs public parameters pp composed of the (q,F).

$Eval\left(\left(q,F\right),x^{\prime }\right)\to \left(y^{\prime },\pi \right)$ is a slow cryptographic algorithm that takes the (q,F) and a random seed $x^{\prime }\in X\subseteq F_{q}^{m}$ as input and compute a $y^{\prime }\in Y$ together with a possibly empty proof π.

$Verify(\left(q,F\right),x^{\prime },y^{\prime },\pi )\to \left\{accept,reject\right\}$ is a deterministic cryptographic algorithm, in which the algorithm inputs (q,F), random seed $x^{\prime }$, outputs $y^{\prime }$ and proof π. If $F\left(x^{\prime }\right)=y^{\prime }$, output accept; Otherwise output reject.

In order to ensure that the solution $y^{\prime }$ is unique, *F* is required to be injective on *Y*.

### 2.2. Incremental Verifiable Delay Functions

**Definition** **2.**
*(Incremental verifiable delay functions.) The system V = (Setup, Eval, Verify) is a incremental verifiable delay function if the time parameter T of the verifiable delay function is not uniquely determined and is allowed to be determined in the output π of Eval, which does not generate additional proofs.*


Since the verifiable delay function is a sequential function [11], Boneh et. al. propose the use of tight incremental verifiable computation to construct an incremental verifiable delay function construction. Next, here’s how to build an incremental verifiable delay function with a tight incremental verifiable computation.

Incremental verifiable computation was first proposed by Valiant [12]. After that, Bitansky et al. [13] applied it to distributed computions and to other proof systems. The incremental verifiable computation is to guarantee that the prover can generate a proof that a certain state is indeed the current state of the computation at every incremental step of the computation. The proof is updated after every step of the computation to produce a new proof. Iterative sequence functions can be implemented via tight incremental verifiable computation, which captures the primitives required by verifiable delay functions.

Let $f_{\lambda }:N\times X\to X$ be an interated sequential function with round function $g_{\lambda }$ having (T,ϵ)-sequentiality. An incremental verifiable computation system for an interated sequential function $f_{\lambda }$ is polynomial time algorithm (IVCGen, IVCProve, IVCVerify) that satisfy completeness, succinctness and soundness.

Completeness.
(3)$$\forall x\in X,\mathrm{Pr}\left[\begin{array}{cc} \mathrm{IVCVerify}(\mathrm{vk},x,y,k,\pi )=\mathrm{Yes}| & \left(\mathrm{vk},\mathrm{ek}\right)\overset{R}{\leftarrow }\mathrm{IVCGen}\left(\lambda ,f\right)\\ | & \left(y,\pi \right)\overset{R}{\leftarrow }\mathrm{IVCProve}\left(\mathrm{ek},k,x\right) \end{array}\right]=1$$

Succinctness. The length of a proof is bounded by ploy(λ,log(kT)).

Soundness. The soundness satisfied by the incremental verifiable computation is sub-exponential soundness. For all algorithm A running in time $2^{o(\lambda )}$.
(4)$$\mathrm{Pr}\left[\begin{array}{cc} \;IVCVerify\;y(vk,x,y,k,\pi )=\mathrm{Yes}\;\;| & \left(\mathrm{vk},\mathrm{ek}\right)\overset{R}{\leftarrow }\mathrm{IVCGen}\left(\lambda ,f\right)\\ f(k,x)\neq y\;\;\;\;\;\;\;\;\;\;\;\;\;\;\;\;\;\;\;\;\;\;\;\;\;\;\;\;\;\;\;\;\;\;\;\;\;\;\;\;\;| & \left(x,y,k,\pi \right)\overset{R}{\leftarrow }A\left(\lambda ,vk,ek\right) \end{array}\right]<\mathrm{negl}\left(\lambda \right)$$

Next, we introduce the verifiable delay function construction based on tight incremental verifiable computation. Let a family $f_{\lambda }$, where each $f_{\lambda }$, $N\times X_{\lambda }\to X_{\lambda }$ is defined by $f_{\lambda }\left(k,x\right)=g_{\lambda }^{k}\left(x\right)$. Here $g_{\lambda }$ is a (*T*, ϵ)-sequential function on an efficiently sampleable domain of size $O(2^{\lambda })$.

Setup(λ,T)→pp=((ek,k),vk) is a randomized algorithm that takes a delay parameter *T* and a security parameter λ as input and outputs public parameters pp composed of the evaluation key ek, a largest integer *k* and the verification key vk. Generate (ek,vk) by running IVCGen(λ, $f_{\lambda }$).

Eval((ek,k),x)→(y,π) is a slow cryptographic algorithm that takes the evaluation key ek, a largest integer *k* and a random seed *x* as input and runs IVCProve(ek, *k*, *x*), and outputs a *y* together with a possibly empty proof π.

Verify(vk,x,y,π)→{accept,reject} is a deterministic cryptographic algorithm, in which the algorithm runs and outputs IVCVerify(vk, *x*, *y*, *k*,π). If $f_{\lambda }\left(x,k\right)=y$, output accept; Otherwise output reject.

Since *T* is fixed in the public parameters pp. However, it is also possible to directly assign the *T* to Eval algorithm. Therefore, a tight incremental verifiable computation based the verifiable delay function is an incremental verifiable delay function.

### 2.3. Decodable Verifiable Delay Functions

**Definition** **3.**
*(Decodable verifiable delay functions.) The system V = (Setup, Eval, Verify) is a decodable verifiable delay function if there is an algorithm in the verifiable delay function that can decode input x backwards from output y. If the decoding is efficient then no additional proof π is required [14].*


Using a slow and easy to verify function with exponentiation [15] based calculations in a finite group can be constructed a decodable verifiable delay function. Boneh et al. propose a simple exponentiation-based decodable verifiable delay functions with bounded pre-computation. However, the adversary cannot run a long pre-computation between the time the public parameter pp is exposed and the time that the verifiable delay function is computed.

Next, we introduce a decodable verifiable delay function based on an exponentiation in a finite group. Let $L=\{\iota_{1},\iota_{2},...,\iota_{T}\}$ be the first *T* odd promes, namely $\iota_{1}=3,\iota_{2}=5,$ etc. Let *P* be the product of the primes in *L*, namely $P:=\iota_{1}\cdot \iota_{2}\cdot ...\cdot \iota_{T}$.

Setup(λ,T,b)→pp=(ek,vk) is a randomized algorithm that takes a delay parameter *T*, a security parameter λ and a preprocessing security parameter *b* as input and outputs public parameters pp composed of the evaluation key ek and the verification key vk.

In algorithm Setup, let a integer module *N* multiplicative group G := (Z/*N*N)${}^{*}$ and a random hash function H:Z→G. The algorithm needs to compute $h_{i}\leftarrow H\left(i\right)\in G$, for $i\in \{1,2,...,b=2^{30}\}$ then compute $g_{i}:=h_{i}^{1/P}$. It outputs the evaluation key ek := (G, *H*, $g_{1}$, $g_{2}$, ..., $g_{b}$) and the verification key vk := (G, *H*).

Eval(ek,x)→y is a slow cryptographic algorithm that takes the evaluation key ek and a random seed *x* as input and outputs a *y*.

In algorithm Eval, using random hash function to map a random seed *x* to a size of λ random subset $L_{x}\subseteq L$ and random subset $S_{x}$ of λ values in $\left\{1,2...,b=2^{30}\right\}$. At the same time, let $P_{x}$ be the product of all prime numbers in $L_{x}$. Let *g* be $g:=\Pi_{i\in S_{x}}g_{i}\in G$ and the seed solution *y* is simply $y\leftarrow g^{P/P_{x}}\in G$.

Verify(vk,x,y,π)→{accept,reject} is a deterministic cryptographic algorithm, in which the algorithm inputs verification key vk, random seed *x*, outputs *y* and proof π. If f(x)=y, output accept; Otherwise output reject.

In algorithm Verify, let *h* be $h:=\Pi_{i\in S_{x}}H\left(i\right)\in G$ and if and only if $y^{P_{x}}=h$∈G, where $P_{x}$ and $S_{x}$ are calculated by the algorithm Eval(ek,x).

The preprocessing parameter *b* in an exponentiation-based the decodable verifiable delay function ensures the security of the construction. The construction requires a trusted setup [16,17], but can be eliminated by choosing a random number large enough.

### 2.4. Trapdoor Verifiable Delay Functions

**Definition** **4.**
*(Trapdoor verifiable delay functions.) The system V = (Keygen, Trapdoor, Eval, Verify) is a trapdoor verifiable delay function if there is a secret key sk that can quickly get the output of Eval through the input of Eval. In other words, the trapdoor verifiable delay function can bypass the delay parameter to quickly calculate the result through the trapdoor [18].*


Given a pair of Alice’s public-secret keys (pk, sk), where pk is Alice’s public key and sk is the secret key. Alice is able to quickly evaluate trap [19,20] functions $Trapdoor_{sk}$ on *x* with a secret key sk. Let *T* be an implicit time function about the security parameter λ and *x* be a piece of data. Except for Alice, everyone else can only compute the public evaluation function $Eval_{pk}$ with the public key pk in *T*-sequential steps and the calculation is slow, but the result between $Eval_{pk}$ and $Trapdoor_{sk}$ is equal. A trapdoor verifiable delay function comsists of four algorihtm (Keygen, Trapdoor, Eval, Verify).

Genkey(λ)→(pk,sk) is a key generation algorithm that takes a security parameter λ as input and outputs Alice’s public key pk and the secret key sk. Meanwhile, Alice’s public key is publicly valid, and the secret key is known only to Alice herself.

$Trapdoor_{sk}\left(x,T\right)\to \left(y,\pi \right)$ is a slow cryptographic algorithm that takes an implicit time function *T* about the security parameter λ and a piece of data *x* as input, and uses the secret key sk to output *y* together with a possibly empty proof π. The function *T* is a sequence of sequential steps required to compute the same output *y* without knowledge of the secret key sk.

$Eval_{pk}\left(x,T\right)\to \left(y^{\prime },\pi^{\prime }\right)$ is a slow cryptographic algorithm to evaluate the function on *x* using only the public key pk. It produces an output *y* associated with $y^{\prime }$ and a possibly empty proof $\pi^{\prime }$. This procedure is meant to be infeasible in time less than *T* (this will be expressed precisely in the security requirements).

Verify(x,T,y,π)→{accept,reject} is a deterministic cryptographic algorithm to verify if *y* is indeed the correct output for *x*, associated with the public key pk and the evaluation time *T*, possibly with the help of the proof π.

The time delay *T* is a function of the security parameter λ and *T* is an input to each algorithm, so the security parameter λ is implicitly an input to each of these procedures. Generate a public-secret key pair (pk,sk) through the key generation algorithm Genkey. Given a piece of data *x* and time delay parameter *T*, let $Trapdoor_{sk}\left(x,T\right)\to \left(y,\pi \right)$ and $Eval_{pk}\left(x,T\right)\to \left(y^{\prime },\pi^{\prime }\right)$. If $y=y^{\prime }$ and Verify(x,T,y,π) = $Verify(x,T,y^{\prime },\pi^{\prime })$ output *accept*; Otherwise output *reject*.

## 3. Verifiable Delay Functions Based on Algebraic Assumptions

### 3.1. Construction Based on Finite Abelian Groups of Unknown Order

Verifiable delay functions can be constructed by showing Rivests-Shamir-Wagner (RSW) when the time-lock puzzle [21,22] is publicly verifiable. To be precise, giving a statistically sound public-coin protocol [23] to prove that a tuple (T,N,x,y) satisfies $y=x^{t}\left(modN\right)$ verifiers do not know the decomposition of *N* and its running time is mainly to solve the puzzle, where the time $t=2^{T}$ is a power of two. This construction solves an instance of the time-lock puzzle, and computes a proof of correctness, which allows anyone to efficiently verify the result.

Pietrzak [24] proposed a verifiable delay function based on finite abelian groups [25,26] of unknown order consisting of four algorithm (Setup,Genkey,Sloth,Verify).

$Setup(1^{\lambda })\to N$ inputs the statistical security parameter $1^{\lambda }$ output *N*, where the λ defines another security parameter $\lambda_{RSA}$ specifying the bitlength of an $\lambda_{RSA}$ modulus and *N* is the single $\lambda_{RSA}$ bit RSA modulus of public parameters. The N:=p·q is composed of two $\lambda_{RSA}/2$-bit secure prime numbers *p* and *q* randomly selected by the Setup algorithm.

Genkey(N,T)→(x,T) samples a random number $x\in QR^{+}$ and outputs (x,T).

Define $QR_{N}\overset{\mathrm{def}}{=}\left\{z^{2}modN:z\in Z_{N}^{*}\right\}$ as quadratic residues and the signed quadratic residues [27] are the group $QR^{+}\overset{\mathrm{def}}{=}\{|x|:x\in QR_{N}\}$. In a verifiable delay function, calculating $x^{2^{T}}$ is difficult in $\left(Z_{N}^{*},\cdot \right)$. Pietrzak uses $\left(QR_{N}^{+},\circ \right)$ instead of $\left(Z_{N}^{*},\cdot \right)$. Because $|QR_{N}\left|=\right|Z_{N}^{*}|/4$, the probability that a random number in ϵ is also in $QR_{N}$ is 1/4. Therefore, if one can break the assumption with probability ϵ over $QR_{N}$, the assumption can also be broken over $Z_{N}^{*}$ with probability ϵ/4. Then, they uses $\left(QR_{N},\cdot \right)$ instead of $\left(QR_{N}^{+},\circ \right)$ in the proof. This approach can make the proof more efficient because the multiplication mod *N* in $QR_{N}$ is more convenient and simpler than the ∘ operation in $QR_{N}^{+}$. Since $\left(QR_{N},\cdot \right)$ and $\left(QR^{+},\circ \right)$ are isomorphic, it is proved $\left(QR_{N},\cdot \right)$ means $\left(QR^{+},\circ \right)$ has the same security.

Let random number $x\in QR_{N}$ and $y=x^{2^{T}}modN$ in $\left(QR_{N},\cdot \right)$, and $x^{\prime }=\left|x\right|$ and $y^{\prime }=(x^{\prime }=\left|y\right|$ in $\left(QR_{N}^{+},\circ \right)$, where $y^{\prime }=\left|y\right|$ and $y=|y^{\prime }{|}^{-1}$, and $y\in y^{\prime },N-y^{\prime }$. Although it is not certain whether $y=y^{\prime }$ or $y=N-y^{\prime }$, *y* has a 1/2 probability of getting the correct value. This shows that given an algorithm that calculates $x^{2^{T}}$ in $QR_{N}^{+}$ with probability ϵ in time *T*, it is possible to obtain an algorithm that calculates $x^{2^{T}}$ in $QR_{N}^{+}$ in time when time *T* and probability ϵ/2 are essentially the same.

Sloth(N,(x,T))→(y,π) is a slow algorithm that takes the *N* and a random seed *x* and time delay parameter *T* as input and outputs a *y* together with π, where $y=x^{2^{T}}$ is the solution of the RSW time-lock puzzle in $QR_{N}^{+}$ and $\pi ={\left\{u_{i}\right\}}_{i}\in \left[T\right]$ is a possibly empty proof that *y* has been correctly evaluated. It is derived by applying the Fiat-Shamir heuristic [28,29] to the protocol.

Verify(N,(x,T),(y,π))→{accept,reject} is a deterministic cryptographic algorithm to verify if *x*, *y* and all $u_{i}$ are in $QR_{N}^{+}$. If these are not the case output reject. Otherwise, all $x_{i}$ and $y_{i}$ should be calculated, and $y_{T+1}\overset{?}{=}x_{T+1}^{2}$ should be judged. If all the above are satisfied, output accept.

### 3.2. Construction Based on Elliptic Curve Cryptography

De Feo et al. [30] designed a new verifiable delay function using isogenic and bilinear pairs [31,32] of super-singular elliptic curves [33], and this framework is non-interactive in nature, the output can be effectively verified without additional proofs. Before describing this structure, let’s introduce some basic factors of super-singular curves, pairings and isogenies.

Elliptic curves on finite fields are described in detail in [34,35,36] and their use in cryptography is described in detail in [37,38,39]. In addition, the ideal class group of quadratic imaginary fields are explained in [40] and the maximal orders of quaternion algebras are introduced in [41,42].

Let *C* be an elliptic curve defined over a finite field $F_{q}$ characterized by *p* and the order of $C(F_{q})$ is $\#C(F_{q})=q+1-L$, where *L* is the trace of the Frobenius endomorphism π. If and only when *p* divided *L*, the curve can be called a super-singular elliptic curve. Each super-singular curve is isomorphic to the curve defined on $F_{p^{2}}$, and for the fixed prime number *p*, there are only a finite number of super-singular curves until isomorphism.

Weil pairing [43] $e_{N}:C\left[N\right]\times C\left[N\right]\to \mu_{N}$ with bilinear pairs is defined on super-singular curves are used to describe verifiable delay functions. That pairing needs to be satisfied the compatibility condition $e_{N}\left(\varphi \left(P\right),Q\right)=e_{N}\left(P,\tilde{\varphi }\left(Q\right)\right)$ for any isogeny $\varphi :C\to C^{\prime }$ and points $P\in C\left[N\right],Q\in C^{\prime }\left[N\right]$.

Verifiable delay functions from super-singular curves. Let $X_{1},X_{2},Y_{1},Y_{2}$, G be groups of prime order *N*. Let $e_{X}:X_{1}\times X_{2}\to G$ and $e_{Y}:Y_{1}\times Y_{2}\to G$ be non degenerate bilinear pairings, where a pair of bijections $\varphi :X_{1}\to Y_{1}$ and $\tilde{\varphi }:Y_{2}\to X_{2}$ and φ and $\tilde{\varphi }$ are group isomorphisms. Let g be any generator of $X_{1}$ and $\left(N,X_{1},X_{2},Y_{1},Y_{2},G,e_{X},e_{Y},g,\varphi \left(g\right)\right)$ be the public parameters. The verifiable delay function is the map $\tilde{\varphi }$ and the maps $\varphi ,\tilde{\varphi }$ are also part of the public parameters. To verify the output, one checks that $e_{X}\left(g,\tilde{\varphi }\left(Q\right)\right)=e_{Y}\left(\varphi \left(g\right),Q\right)$, where $Q\in Y_{1}$ is the point at which Eval calculates the value.

De Feo et al. propose verifiable delay functions for super-singular curves over prime field $F_{p}$ and $F_{p^{2}}$, using super-singular elliptic curves for the pairing groups, and isogenies of prime power degree for the maps φ, $\tilde{\varphi }$. Next, we mainly introduce verifiable delay functions with super-singular curves over a prime field $F_{p}$.

Let *p* be prime so that p+1 contains the larger prime factor *N*. Let degree l=2, p=7mod8 or a small prime such that $(\frac{-p}{l})=1$. Take the super-singular elliptic curve $C/F_{p}$, and denote by $e_{N}\left(\cdot ,\cdot \right)$ the Weil pairing on C[N].

$Setup\left(\lambda ,T\right)\to \left(ek,vk\right)=(\tilde{\varphi },\left(C,C^{\prime },g,\varphi \left(g\right)\right))$ is a randomized algorithm that takes a delay parameter *T* and a security parameter λ as input. First, a super-singular curve $C/F_{p}$ needs to be chosen and a direction needs to be chosen on the horizontal *l*-isogeny graph to compute a cyclic isogeny $\varphi :C\to C^{\prime }$ of degree $l^{T}$ and its dual $\tilde{\varphi }$ in this algorithm. Next, the algorithm chooses a generator g of $X_{1}=v^{-1}\left(\tilde{C}\left[N\right]\cap \tilde{C}\left(F_{p}\right)\right))$ and compute φ(g), where $\mu \in F_{p}$ is a non quadratic residue and $\tilde{C}$ is a quadratic twist of *C*. Finally, the algorithm outputs public parameters $\left(ek,vk\right)=(\tilde{\varphi },\left(C,C^{\prime },g,\varphi \left(g\right)\right))$, where (ek,vk) composed of the evaluation key ek and the verification key vk, and $\left(\tilde{\varphi },\left(C,C’,g,\varphi \left(g\right)\right)\right)$ composed of the map $\tilde{\varphi }$, the cyclic isogeny $C\to C^{\prime }$, generator of $X_{1}$ and the map φ(g) of generator g.

$Eval\left(\tilde{\varphi },Q\in Y_{1}\right)\to \tilde{\varphi }\left(Q\right)$ is a slow cryptographic algorithm that takes the map $\tilde{\varphi }$ and a point $Q\in Y_{1}$ as input and outputs $\tilde{\varphi }\left(Q\right)$.

$Verify(C,C^{\prime },g,Q,\varphi \left(g\right),\tilde{\varphi }\left(Q\right))\to \left\{accept,reject\right\}$ is a deterministic cryptographic algorithm. The algorithm needs to verify that $\tilde{\varphi }\left(Q\right)\in X_{2}=C\left[N\right]\cap C\left(F_{p}\right)$ and $e_{N}\left(g,\tilde{\varphi }\left(Q\right)\right)=e_{N}\left(\varphi \left(g\right),Q\right)$. If all the above are satisfied, output accept; Otherwise, output reject.

## 4. Verifiable Delay Functions Based on Structural Assumptions

Ephraim et al. design continuous verifiable delay functions based on a high arity tree [44], where each intermediate state of the computation can be verified and proofs of the computation can be efficiently merged. It is a verifiable delay function based on the assumption of tree structure constructed on the basis of the repeated square [45] assumption. The continuous verifiable delay function only depends on the Fiat-Shamir heuristic for a constant round proof system. Next, we introduce continuous verifiable delay functions based on high arity trees.

First, the computational steps correspond to a specific traversal of a (k+1)-ary tree of height $h=log_{k}^{B}$. Each node in the (k+1)-ary tree is related to a statement (x,y,T,π) of the underlying verifiable delay functions, where the output value $y=x^{2^{T}}$ and the possible empty proof π are the corresponding proofs of correctness. If *x* is the node’s input, the difficulty $T=k^{h-l}$ is determined by its height in (k+1)-ary tree and *l* is root node.

Next, they define a tree structure. Starting from the root node with input $x_{0}$ and difficulty $T=k^{h}$, divide the tree structure into *k* segments $x_{1},x_{2},...,x_{k}$ similar to the verifiable delay functions structure. In a tree-based verifiable delay functions structure, only calculating the input *x* of a leaf node from the previous state can guarantee that each step of calculation corresponds to the calculation of a single leaf’s statement.

Before introducing continuous verifiable delay functions, let’s review unique verifiable delay functions. Next, the interactive proof that language $L_{N,B}$ corresponding to repeated squares is transformed into unique verifiable delay functions by using the Fiat-Shamir heuristic, where
(5)$$L_{N,B}=\left\{\left(x_{0},y,T\right):\begin{array}{c} \left.y^{2}={\left(x_{0}\right)}^{2^{T+1}}\;\mathrm{mod}\;N,x_{0}\;\mathrm{is}\;\mathrm{valid}\;\mathrm{and}\;T\leq B_{3}\right\}\\ y=⊥,\mathrm{othersize}\; \end{array}\right.$$

A unique verifiable delay function is composed of the following four algorithm (uVDF.Gen, uVDF.Sample, uVDF.Eval, uVDF.Verify).

$uVDF.Gen\left(1^{\lambda }\right)\to pp=\left(N,B,k,d,hash\right)$ is a randomized algorithm that takes a statistical security parameter $1^{\lambda }$ as input and outputs public parameters pp composed of the RSA modulus *N*, the upper bound *B*, the arity *k*, a constant *d* and a hash function hash, where hash←H, k=λ and B=B(λ).

$uVDF.Sample\left(1^{\lambda },pp\right)\to x_{0}$ takes a statistical security parameter $1^{\lambda }$ and the public parameters pp and sample and output a random element $x_{0}\leftarrow Z_{N}^{*}$ such that $gcd(x_{0}\pm 1,N)=1$ and $x_{0}=\left|x_{0}\right|$.

$uVDF.Eval\left(1^{\lambda },pp,\left(x_{0},T\right)\right)\to \left(y,\pi \right)$ is a slow cryptographic algorithm that takes a statistical security parameter $1^{\lambda }$, the public parameters pp, a random element $x_{0}$ and the time delay parameter *T* as input and outputs a *y* together with a possibly empty proof π.

$uVDF.Verify\left(1^{\lambda },pp,\left(x_{0},T\right),\left(y,\pi \right)\right)\to \left\{0,FS-Verify\left(pp,\left(x_{0},T\right),\left(y,\pi \right)\right)\right\}$ is a deterministic cryptographic algorithm. If all the above are satisfied, output accept. Otherwise, output $FS-Verify(pp,\left(x_{0},T\right),\left(y,\pi \right))$. The FS−Verify is a verification algorithm for Fiat-Shamir transformations defined on the protocol for language $L_{N,B}$ relative to some hash family H. For details of the algorithm, see [46]. Then, we review the definition of a continuous verifiable delay function and describe it in detail.

**Definition** **5.**
*(Continuous verifiable delay functions.) Let B,l:N→N and ϵ∈(0,1). A (B,l,ϵ)-continuous verifiable delay function is a tuple (cVDF.Gen, cVDF.Sample, cVDF.Eval, cVDF.verify) such that (cVDF.Gen, cVDF.Sample, cVDF.Eval) is a (a, B, l, ϵ)-iteratively sequential function, (cVDF.Eval,cVDF.verify) is a B-sound function.*


At a high level, the continuous verifiable delay function will iteratively compute each leaf node in a $\left(pp_{uVDF},d^{\prime },g\right)$-puzzle tree, where $pp_{uVDF}$ = (*N*, *B*, *k*, *d*, hash) are the public parameters of the underlying unique verifiable delay function and *g* is the starting point of the tree given by uVDF.Sample.

Next, we define a frontier. At a high level, for a leaf *s*, the frontier of *s* will correspond to the state of the continuous verifiable delay function upon reaching *s*. Specifically, it will contain all nodes whose values have been computed at that point, but whose parents’ values have not yet been computed.

**Definition** **6.**
*(Frontier.) For a node s in a ($pp_{uVDF},d^{\prime },g)$-puzzle tree, the frontier of s, denoted frontier(s), is the set of pairs $\left(s^{\prime },val\left(s^{\prime }\right)\right)$ for nodes $s^{\prime }$ that are left siblings of any of the ancestors of s. We note that s is included as one of its ancestors.*


Next, we review the formal details of continuous verifiable delay functions, which is a tuple (cVDF.Gen, cVDF.Sample, cVDF.Eval, cVDF.Verify).

$cVDF.Gen\left(1^{\lambda }\right)\to pp=\left(pp_{uVDF},d^{\prime },hgt\right)$ is a randomized algorithm that takes a statistical security parameter $1^{\lambda }$ as input and outputs public parameters pp composed of the $pp_{uVDF}=\left(N,B,k,d,hash\right)$, a constant $d^{\prime }$ and a tree height $hgt=\left\lceil log_{k}\left(B\right)\right\rceil -d^{\prime }$.

$cVDF.Sample\left(1^{\lambda },pp\right)\to v=\left(g,0^{h},\phi \right)$ takes a statistical security parameter $1^{\lambda }$ and public parameters pp as input and outputs a random element *v*, where *g*←uVDF.Sample($1^{\lambda }$, $pp_{uVDF}$) is sampled by the Sample algorithm of unique verifiable delay functions.

$cVDF.Eval\left(1^{\lambda },pp,v\right)\to v^{\prime }$ takes a state *v* corresponding to a leaf *s* in the tree and computes the next state $v^{\prime }$ corresponding to the next leaf. Each state *v* will be of the form (g,s,F), where *s* is the current leaf in the tree, *F* is the frontier of *s* and *g* is the starting point of the tree given by uVDF.Sample.

$cVDF.Verify(1^{\lambda },pp,\left(v,T\right),v^{\prime })\to \left\{accept,reject\right\}$ verifies the state *v* by recursively running this verification algorithm and whether $v^{\prime }$ is correct. Output accept if all the check conditions of the continuous verifiable delay function are satisfied; Otherwise output reject.

The heart of our construction is the cVDF.Eval functionality which takes a state *v* corresponding to a leaf *s* in the tree and computes the next state $v^{\prime }$ corresponding to the next leaf. Each state *v* will be of the form (g,s,F), where *s* is the current leaf in the tree and *F* is the frontier of *s*. Then, cVDF.Eval($1^{\lambda }$, pp, (g,s,frontier(s)) will output (*g*, s+1, frontier(s+1)). There are three phases of the algorithm cVDF.Eval. First, it checks that its input is well-formed. It then computes val(s) using frontier(s), and then computes frontier(s+1) using both frontier(s) and val(s).

## 5. Post-Quantum Verifiable Delay Functions

In 2021, Jorge Chavez-Saab et al. [47] researched the problem of constructing post-quantum secure verifiable delay functions, especially verifiable delay functions based on super-singular isogeny. They propose an arithmetic structure specifically for homologous settings using succinct non-interacting arguments (SNARGs) [48] to achieve good asymptotic efficiency. This isogeny-based verifiable delay functions has the advantages of post-quantum security [49], quasi-logarithmic verification, and does not require a trusted setup. Since the Eval algorithm for postquantum verifiable delay functions involves isogeny walks on super-singular elliptic curves that can be publicly verified through a SNARG-based verification process. Formally, a verifiable delay function is composed of three main algorithm:

Setup(λ,T)→pp=(ek,vk) takes a delay parameter *T* and a security parameter λ as input and outputs public parameters pp composed of the evaluation key ek and the verification key vk.

Eval(ek,x)→(y,π) takes the evaluation key ek and a certain input *x* as input and calculates an output *y* and a possibly empty proof π.

The function involves computing walks of length *T* in the 2-isogeny graph of super-singular curves on $F_{p^{2}}$, where $p^{2}\equiv 9mod16$ (which is required to apply Kong’s square root algorithm [50]) and p=poly(T). Given a time delay parameter *T*, and the evaluator needs to compute a walk of length *T* on the 2-isogeny graph, where the exact path is determined by a string *s* and the path is not backtracking. Given the two *v*-invariant curves $v_{i}$ and $v_{i+1}$, they are 2-isogenous over $F_{p^{2}}$ if and only if the modular polynomial $\Phi (v_{i},v_{i+1})$ vanishes. For a fixed $v_{i}$ the next curve in the path can be calculated by finding the root of $\Phi (v_{i},A)$. If $A=v_{i-1}$ is a known root of Φ(A) = $A^{3}$ + $aA^{2}$ + bA + *c* then Φ(A) can rewrite Φ(A) = ($A-v_{i-1}$)($A^{2}$+ (*a* + $v_{i-1}$)*A* + *b* + $av_{i-1}$ + $v_{i-1}^{2}$) and focus on finding the roots of the quadratic factor. After the square root is calculated, the evaluator selects the symbol using the input string, resulting in a definite traversal process.

Verify(vk,x,y,π)→{accept,reject} inputs verification key vk, a certain input *x*, a output *y* and a proof π. If f(x)=y, output accept; Otherwise output reject.

Since the postquantum verifiable delay functions is constructed over SNARG, a deterministic process and a fixed symbol are required for the SNARG verification process. For the validation process, the evaluator keeps track the results of the evaluation and construct an SNARG, and the values resulting from the evaluation process must be satisfied.

De-Feo et al. proposed a new verifiable delay function framework based on the assumption of elliptic curve cryptography, and instantiated this framework using super-singular elliptic curves and bilinear pairs. The structure of this verifiable delay function is non-interactive in nature, and the output can be effectively verified without additional proofs. However, the only secure way to instantiate a verifiable delay function requires a trusted setup to perform a random isogeny traversal. In fact, this setup needs to start with super-singular elliptic curves with unknown autohomomorphic rings. In order to explain how to implement the proposed verifiable delay function on elliptic curves with commutative self-homomorphic rings, Shani later used the idea of verifiable delay functions based on isogeny and pairing proposed by De-Feo et al. to construct developed a verifiable delay function based on isogeny without pairing. The scheme is a combination of a time-lock puzzle and a trapdoor verifiable delay function.

However, neither scheme is quantum secure. Thus, Chavez-Saab et al. studied the problem of constructing post-quantum secure verifiable delay functions, especially verifiable delay functions based on super-singular isogeny. They propose an arithmetic structure specifically for homologous settings using SNARGs to achieve good asymptotic efficiency. This isogeny-based verifiable delay function has the advantages of post-quantum security, quasi-logarithmic verification, and does not require a trusted setup.

This verifiable delay function construction finds codomain curves, which are computed from any three-point evaluation, so the problem in the verifiable delay function setting could be regarded as a general problem. The precomputation time allowed in the setting is given before learning the isogeny to be evaluated, suggesting that this verifiable delay function construction uses a different isogeny for each input. This verifiable delay function relies on a very weak assumption, so it is more secure. Starting from a public curve means we do not need a trusted setup. Next, we analyze two security properties of post-quantum verifiable delay functions based on super-singular elliptic curves: sequentiality and soundness.

Sequentiality. Any protocol that does not prescribe isogeny walk in some way is insecure in terms of sequentiality. The evaluator can be asked to provide SNARG proof of any large degree of isogeny and consider this to be a good proof of sequentiality even if the output is not unique. However, if the evaluator is free to choose the path, this does not constitute a proof of sequential computation. The sequentiality of post-quantum verifiable delay functions relies on isogeny shortcut problem of De Feo et al.. If no pair of random algorithms $A_{0}$ (running in total time poly(T,λ)) and $A_{1}$ (running in parallel time less than *T*) can win the following sequential game with non-negligible probability, then the post-quantum verifiable delay functions based on super-singular elliptic curves is sequential. The Setup must use secret randomness to choose the starting curve, and the Setup is left with choosing isogeny and generators, both of which can use public randomness. Furthermore, $A_{0}$ allows for poly(T) computations, so it can compute isogeny on generators. Therefore, setting aside the choice of starting curve, the Setup can be absorbed into $A_{0}$, which proves that the verifiable delay function of the post-quantum is sequential.

Soundness. The soundness of post-quantum verifiable delay functions based on super-singular elliptic curves completely depends on the SNARG proofs. Succinctness and non-interactiveness are achieved by generating SNARG through a transform that acts on any probabilistic checkable proofs. As long as the hash function is collision-resistant, it is straightforward to prove that the soundness of the structure reduces to the soundness of the original probabilistic checkable proofs. Therefore, the post-quantum verifiable delay functions construction based on super-singular elliptic curves is sufficiently soundness.

## 6. Verifiable Delay Functions Applications

With the proposal of more and more verifiable delay function schemes, the application of verifiable delay function in distributed systems is also well known. Next, as shown in Figure 2, this paper introduces several particularly important applications. In Table 3, the blockchain-related applications of verifiable delay functions are described.

Computational Timestamping. Almost all know proof-of-stake [51] systems face the problem of long-range attack [52]. In a proof-of-stake protocol, a group of stakeholders has voting rights proportional to their stake at any time. Assume that the majority of stakeholders have no reason to break the system, since the stakeholders themselves are incentivized. However, when too many stakeholders sell their stake, they can collude to create a new longer system to replace history. An external timestamp mechanism [53] can prove to the user that the true history of the system is much older.

Incremental verifiable delay functions can provide computational evidence that a given version of the state system is older by proving that long-running verifiable delay function computations has been performed on the true history after divergence from the fraud history. This has the potential to detect long-range attack without relying on external timestamping mechanisms. In 2019, Landerreche et al. [54] presented the first treatment of non-interactive publicly verifiable timestamping schemes and a secure timestamping scheme based on a verifiable delay function is proved. The timestamping scheme [55] consists of sequence of verifiable delay function proofs linked to each other by a cryptographic hash function, modeled as a sequence of random oracles. Add verifiable delay functions to the sequence to increase the structure, thus preserving the safety guarantees of the structure.

Public Randomness Beacons. Verifiable delay functions are useful for methods of obtaining random numbers from public sources. For example, constructing public randomness beacons [56] from stock prices, election audit or proof-of-work blockchains [57]. In the stock market, a strong enough stock trader can change the random output of stock prices by influencing the market trend, which greatly affects the fairness of the stock market [58]. The verifiable delay function can increase the security of public verifiable nonces by adding a long enough delay to calculate the beacon, which helps ensure that malicious traders do not have enough time to adjust the market. In a proof-of-work blockchain’s computational puzzle solving, miners continuously mine and publish puzzles for monetary rewards. However, similarly powerful enough stock traders may manipulate stock prices, sufficiently powerful miners may manipulate beacon results by refusing to publish blocks [59] that produce unfavorable beacon outputs.

In addition, the verifiable delay function can also add some random beacon schemes involving multiple parties. For example, in “commit-and-reveal” [60], the attacker can wait until the end of the reveal phase to decide whether to reveal his or her commitment. If the following three conditions are met: discard the commit phase; integrate everyone’s input at the end of the protocol and put it into the verifiable delay function instead of directly as the result of a random number; set the time parameter T long enough and later than the deadline for the last submission; then even the last-minute submitter has no way of knowing the result of the random number.

Resource Efficient Blockchains. When the blockchain [61] is forked, the consensus participants will choose to mortgage assets on different forked chains to participate in the block generation for their own interests, so that the forked chain may always exist and there will be more and more forks. Seriously endanger the consistency of the system and the consumption of resources. However, resource-efficient mining suffers from nothing-at-stake attacks. Intuitively, since mining is not computationally expensive, miners can easily try to produce many individual forks.

To prevent nothing-at-stake attacks [62], random beacons are used to select a new leader at intervals. At the same time, a verifiable delay function can also be used to increase the security of random beacons in the consensus protocol that uses random beacons to select a new leader. Most of the random number schemes used by these protocols remain secure only when there is a majority of honest participants. Utilizing a verifiable delay function can improve the participation of any honest party.

In addition to the electoral scheme, Cohen proposes to combine proof of resources [63] with an incremental verifiable delay function and use the product of proven resources and induced delay as a measure of blockchain [64,65] quality. This scheme prevents nothing-at-stake attacks by simulating the proof-of-work mining process. The timing of mining blocks is unpredictable, and each miner competes with each other to be the first to mine a block. The difference from the proof-of-work [66] is that this scheme does not actually need to consume too much time resources for parallel computing, and only has certain space resources when entering mining.

Proof of Data Replication. Proof of data replication [67,68,69] is a special type of proof of storage of data that allows a client to verify that it has a unique replica of some data stored on an untrusted server, even if the data is readily available from a public storage source. Proof of data replication is meant to prove that the server has a replica of the data, not that it owns the data. Boneh et al. proposed to provide a publicly verifiable solution using a decodable verifiable delay function that is asymmetric in time. The decodable verifiable delay function prevents the server from dynamically computing the client’s replica when challenged to prove that it correctly stored a replica of the data.

Gritti proposes a publicly verifiable proof of data replication scheme using verifiable delayed functions, and explains the scheme in detail along with a security proof. Given a unique replicator identifier id. Then the server divides the file into *b*-bit sized file blocks Bi and calculate $B_{id}=B_{i}\left(i||id\right)$ where *H* is a collision-resistant hash function $H:\left\{0,1\right\}\to {\left\{0,1\right\}}^{b}$ and i∈{1,...,n}. The output value $y_{i}$ can be obtained by taking the calculated $B_{id}$ as part of the Eval algorithm input of the decodable verifiable delay function. The server stores all $y_{i}$ and the client continuously randomly selects *i* to challenge the server to return $y_{i}$. If the server can respond to the corresponding result to the client within the specified time and the client can obtain $B_{i}$ by decoding $y_{i}$ in a very short time to complete the verification. If the server does not respond to the client correctly, $y_{i}$ must be calculated in *T* steps, but this calculation cannot be completed within the specified time.

Verifiable delay functions are widely used in blockchains. However, verifiable delay functions based on finite groups of unknown order are insecure against an adversary with access to a quantum computer. Quantum computers can easily compute the order of a group using Shor’s algorithm, making it easy to break the application of verifiable delay functions in blockchains. In addition, the post-quantum secure verifiable delay function based on supersingular elliptic curves needs to be verified by SNARG, and the structure is in its infancy and can only be limited to the field of elliptic curves. Therefore, it is an open problem to develop a verifiable delay function that has a simple structure in quantum computers and can be safely applied to blockchain.

## 7. Conclusions and Perspectives

Verifiable delay function is a basic and important tool in the field of cryptography, and it has been widely used in distributed systems. This article firstly introduces the types of verifiable delay functions, and describes the construction schemes of different kinds of verifiable delay functions in detail. Secondly, in the algebraic assumption scheme, verifiable delay function schemes based on finite abelian groups of unknown order and super-singular elliptic curve cryptography scheme are introduced respectively, both of which are not quantum secure. Thirdly, the unique verifiable delay functions and the continuous verifiable delay functions based on tree-structure assumptions are introduced. The continuous verifiable delay function achieves effective verification of the output of each intermediate iteration through continuous iteration, and can efficiently incorporate proofs of final computation. Fourthly, this paper presents a postquantum secure verifiable delay function scheme based on super-singular isogeny, And the security analysis of the verifiable delay function is carried out. Finally, the application of verifiable delay function in different aspects of the blockchain is summarized.

Verifiable delay functions have been extensively studied, and the research results obtained cover various branches of verifiable delay function research. However, the application of the verifiable delay function in the existing research results has not been fully integrated into the blockchain, and the application in the blockchain still needs further research. We hope that this work will stimulate new practical applications of verifiable delay functions in blockchains and continue to investigate theoretically optimal verifiable delay function constructions in the future.

## Figures and Tables

**Figure 1 sensors-22-07524-f001:**
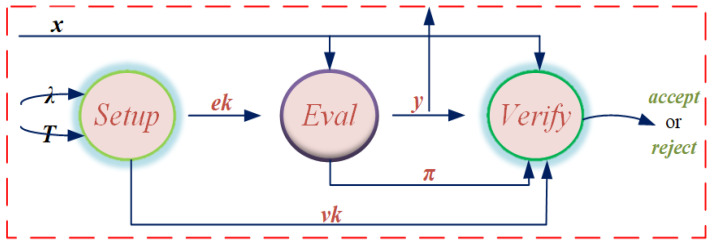
The algorithmic flow of verifiable delay functions.

**Figure 2 sensors-22-07524-f002:**
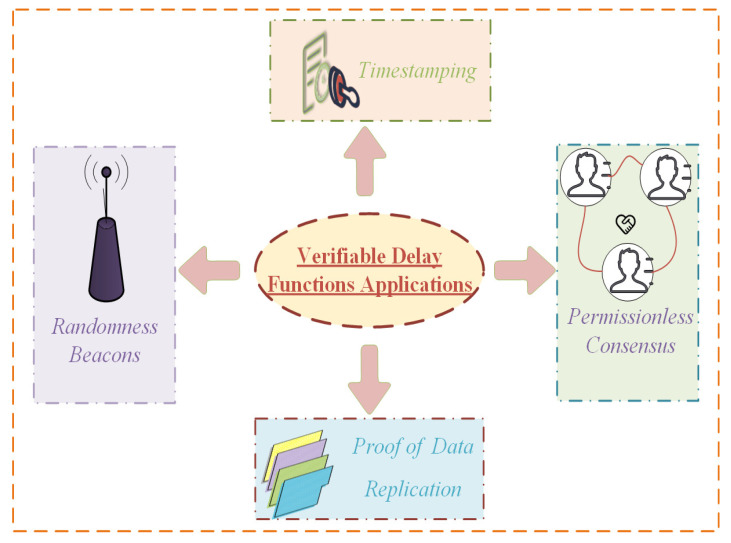
Verifiable delay functions applications.

**Table 1 sensors-22-07524-t001:** Characteristics of verifiable delay functions.

Characteristics	Description
*T*-Sequentiality	The function cannot be calculated in a sequential steps less than *T* to obtain the final result, even given a large amount of parallelism.
Uniqueness	For the input of any verifiable delay functions, only one unique output result shall pass the inspection. Meanwhile, it is necessary to ensure that the probability of the verifier passes the verification because of the proof, but the output result is not the correct result is negligible.
Effective verifiability	The calculation results can be efficiently verified so that the honest party can calculate.

**Table 2 sensors-22-07524-t002:** Classification of verifiable delay functions.

Classification	Description
Weak verifiable delay functions	The function cannot be calculated in a sequential steps less than *T* to obtain the final result, even given a large amount of parallelism.
Incremental verifiable delay functions	All verifiable delay functions need to require the Eval algorithm to be completed in at least *T* steps. If the delay parameter *T* is not uniquely determined in the Setup algorithm, but is allowed to be determined in the Eval algorithm, then the verifiable delay function can be called as incremental verifiable delay function.
Decodable verifiable delay functions	For any verifiable delay function scheme, as long as a random input element *x* can be obtained from the output value *y* in reverse, the verifiable delay function can be called a decodable verifiable delay function. Another output value π for the proof is empty.
Trapdoor verifiable delay functions	If there is an algorithm that enables the party who knows a certain secret key value *sk* to calculate the output value of the verifiable delay function through the *Eval* algorithm too quickly, then the function is a trapdoor verifiable delay function.

**Table 3 sensors-22-07524-t003:** Blockchain-related applications of verifiable delay functions.

Applications	Solution	Purpose
Timestamping	A verifiable delay function is equivalent to a proof of the passage of time, with the input and output of a verifiable delay function on-chain to prove the history of a given block.	Mitigating long-range attacks.
Randomness Beacon	The time delay parameter *T* of the verifiable delay function is set long enough, and the latest block header is used as part of the input in the verifiable delay function, and the final output is the random beacon result.	Enhancing the security of public verifiable random numbers.
Permissionless Consensus	Combine proofs-of-resource with incremental verifiable delay functions and use the product of resource proved and delay induced as a measure of blockchain quality.	Solving nothing-at-stake attacks.
Proof of Replication	Decodable verifiable delay functions generate multiple puzzles, and then the solutions of these puzzles are combined with all replicas to generate new replicas.	Preventing dynamic generation of replicas.

## Data Availability

Not applicable.

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
