# Peer review of "Verifiable Delay Function and Its Blockchain-Related Application: A Survey"

_sensors, 2022, doi:10.3390/s22197524_

Round 1

Reviewer 1 Report

Authors  presented a postquantum secure verifiable delay function scheme based on super-singular homology and summarized the applications of this area. The presented proposal is very interested. However, no promising results, simulation's or analysis were done to verify the concept. Simulation and some analysis would be helpful to add to the paper. Conclusion section needs to be improved as well.

Author Response

Dear Reviewer,

Thanks for your very insightful comment. We have tried our best to address the comments and suggestions from you in our revision.

We are uploading (a) our point-by-point responses to the comments (below), (b) an updated manuscript with blue font indicating changes.

Best regards,

Qiang Wu, Liang Xi, Shiren Wang, Shan Ji*, Shenqing Wang, Yongjun Ren.

Concern # 1: Authors presented a postquantum secure verifiable delay function scheme based on super-singular homology and summarized the applications of this area. The presented proposal is very interested. However, no promising results, simulation's or analysis were done to verify the concept. Simulation and some analysis would be helpful to add to the paper.

Author response: We gratefully thanks for the precious time the reviewer spent making constructive remarks. Theoretical analysis of post-quantum verifiable delay functions are as follows. At the same time, we also added in Section 6.

This verifiable delay function construction finds codomain curves, which are computed from any three-point evaluation, so the problem in the verifiable delay function setting could be regarded as a general problem. And the precomputation time allowed in the setting is given before learning the isogeny to be evaluated, suggesting that this verifiable delay function construction uses a different isogeny for each input. This verifiable delay function relies on a very weak assumption, so it is more secure. Starting from a public curve means we do not need a trusted setup. Next, we analyze two security properties of post-quantum verifiable delay functions based on super-singular elliptic curves: sequentiality and soundness.

Sequentiality. Any protocol that does not prescribe isogeny walk in some way is insecure in terms of sequentiality. The evaluator can be asked to provide SNARG proof of any large degree of isogeny and consider this to be a good proof of sequentiality even if the output is not unique. However, if the evaluator is free to choose the path, this does not constitute a proof of sequential computation. The sequentiality of post-quantum verifiable delay functions relies on isogeny shortcut problem of De Feo et al.. If no pair of random algorithms A1 (running in total time poly(T, λ)) and A1 (running in parallel time less than T) can win the following sequential game with non-negligible probability, then the post-quantum verifiable delay functions based on super-singular elliptic curves is sequential. The Setup must use secret randomness to choose the starting curve, and the Setup is left with choosing isogeny and generators, both of which can use public randomness. Furthermore, A0 allows for poly(T) computations, so it can compute isogeny on generators. Therefore, setting aside the choice of starting curve, the Setup can be absorbed into A0, which proves that the verifiable delay function of the post-quantum is sequential.

Soundness. The soundness of post-quantum verifiable delay functions based on super-singular elliptic curves completely depends on the SNARG proofs. Succinctness and non-interactiveness are achieved by generating SNARG through a transform that acts on any probabilistic checkable proofs. As long as the hash function is collision-resistant, it is straightforward to prove that the soundness of the structure reduces to the soundness of the original probabilistic checkable proofs. Therefore, the post-quantum verifiable delay functions construction based on super-singular elliptic curves is sufficiently soundness.

Concern # 2: Conclusion section needs to be improved as well.

Author response 2: Thanks a lot for your professional comments. Regarding the suggestion about the Conclusion, we have followed your suggestions and have improved the Conclusion section. The details are as follows:

7. Conclusion and Perspectives

Verifiable delay function is a basic and important tool in the field of cryptography, and it has been widely used in distributed systems. This article firstly introduces the types of verifiable delay functions, and describes the construction schemes of different kinds of verifiable delay functions in detail. Secondly, in the algebraic assumption scheme, verifiable delay function schemes based on finite abelian groups of unknown order and super-singular elliptic curve cryptography scheme are introduced respectively, both of which are not quantum secure. Thirdly, the unique verifiable delay functions and the continuous verifiable delay functions based on tree-structure assumptions are introduced. The continuous verifiable delay function achieves effective verification of the output of each intermediate iteration through continuous iteration, and can efficiently incorporate proofs of final computation. Fourthly, this paper presents a postquantum secure verifiable delay function scheme based on super-singular isogeny, And the security analysis of the verifiable delay function is carried out. Finally, the application of verifiable delay function in different aspects of the blockchain is summarized.

Verifiable delay functions have been extensively studied, and the research results obtained cover various branches of verifiable delay function research. However, the application of the verifiable delay function in the existing research results has not been fully integrated into the blockchain, and the application in the blockchain still needs further research. We hope that this work will stimulate new practical applications of verifiable delay functions in blockchains and continue to investigate theoretically optimal verifiable delay function constructions in the future.

Reviewer 2 Report

The article presents a Verifiable delay function and it's an important tool in the field of cryptography, and it has been widely used in distributed systems. Authors introduced the types of verifiable delay functions. Finally, the application of verifiable delay function in different aspects of the blockchain is summarized. I have few suggestions for improvement.

1. Verifiable Delay Functions Applications related with blockchain must be presented in detail. For example, summary of blockchain applications must be tabulated with it's mechanism, types, solution and purpose.

2.  Security analysis and soundness proofs may be included for Post-quantum Verifiable Delay Functions.

3.  Recent literature on blockchain may be considered to enrich the manuscript. Blockchain-based decentralized user authentication scheme for letter of guarantee in financial contract management.

4..  In a survey paper, the open challenges and future directions of Verifiable Delay Function and Its Blockchain-Related Application should be explained in detail as a separate section.

Author Response

Dear Reviewer,

Thanks for your very insightful comment. We have tried our best to address the comments and suggestions from you in our revision.

We are uploading (a) our point-by-point responses to the comments (below), (b) an updated manuscript with blue highlighting indicating changes.

Best regards,

Qiang Wu, Liang Xi, Shiren Wang, Shan Ji*, Shenqing Wang, Yongjun Ren.

Concern # 1: Verifiable Delay Functions Applications related with blockchain must be presented in detail. For example, summary of blockchain applications must be tabulated with it's mechanism, types, solution and purpose.

Author response: Thanks for your valuable comment. In Section 6, we improve the article by adding a table of blockchain-related applications of verifiable delay functions. The table is shown below.

Table 3. Blockchain-related applications of verifiable delay functions.

Applications

Solution

Purpose

Timestamping

A verifiable delay function is equivalent to a proof of the passage of time, with the input and output of a verifiable delay function on-chain to prove the history of a given block.

Mitigating long-range attacks.

Randomness Beacon

The time delay parameter T of the verifiable delay function is set long enough, and the latest block header is used as part of the input in the verifiable delay function, and the final output is the random beacon result.

Enhancing the security of public verifiable random numbers.

Permissionless Consensus

Combine proofs-of-resource with incremental verifiable delay functions and use the product of resource proved and delay induced as a measure of blockchain quality.

Solving nothing-at-stake attacks.

Proof of Replication

Decodable verifiable delay functions generate multiple puzzles, and then the solutions of these puzzles are combined with all replicas to generate new replicas.

Preventing dynamic generation of replicas.

Concern # 2: Security analysis and soundness proofs may be included for Post-quantum Verifiable Delay Functions.

Author response: We gratefully thanks for the precious time the reviewer spent making constructive remarks. Security analysis and soundness proofs of post-quantum verifiable delay functions are as follows. At the same time we also added in Section 5.

This verifiable delay function construction finds codomain curves, which are computed from any three-point evaluation, so the problem in the verifiable delay function setting could be regarded as a general problem. And the precomputation time allowed in the setting is given before learning the isogeny to be evaluated, suggesting that this verifiable delay function construction uses a different isogeny for each input. This verifiable delay function relies on a very weak assumption, so it is more secure. Starting from a public curve means we do not need a trusted setup. Next, we analyze two security properties of post-quantum verifiable delay functions based on super-singular elliptic curves: sequentiality and soundness.

Sequentiality. Any protocol that does not prescribe isogeny walk in some way is insecure in terms of sequentiality. The evaluator can be asked to provide SNARG proof of any large degree of isogeny and consider this to be a good proof of sequentiality even if the output is not unique. However, if the evaluator is free to choose the path, this does not constitute a proof of sequential computation. The sequentiality of post-quantum verifiable delay functions relies on isogeny shortcut problem of De Feo et al.. If no pair of random algorithms A1 (running in total time poly(T, λ)) and A1 (running in parallel time less than T) can win the following sequential game with non-negligible probability, then the post-quantum verifiable delay functions based on super-singular elliptic curves is sequential. The Setup must use secret randomness to choose the starting curve, and the Setup is left with choosing isogeny and generators, both of which can use public randomness. Furthermore, A0 allows for poly(T) computations, so it can compute isogeny on generators. Therefore, setting aside the choice of starting curve, the Setup can be absorbed into A0, which proves that the verifiable delay function of the post-quantum is sequential.

Soundness. The soundness of post-quantum verifiable delay functions based on super-singular elliptic curves completely depends on the SNARG proofs. Succinctness and non-interactiveness are achieved by generating SNARG through a transform that acts on any probabilistic checkable proofs. As long as the hash function is collision-resistant, it is straightforward to prove that the soundness of the structure reduces to the soundness of the original probabilistic checkable proofs. Therefore, the post-quantum verifiable delay functions construction based on super-singular elliptic curves is sufficiently soundness.

Concern # 3: Recent literature on blockchain may be considered to enrich the manuscript. Blockchain based decentralized user authentication scheme for letter of guarantee in financial contract management.

Author response: We appreciate this comment. We have added a reference to the article.

[65] Sasikumar, A.; Karthikeyan, B.; Arunkumar, S.; Saravanan, P.; Subramaniyaswamy, V.; Ravi, L. Blockchain-based decentralized user authentication scheme for letter of guarantee in financial contract management. Malays. J. Comput. Sci. 2022, 1, 62–73.

Concern # 4: In a survey paper, the open challenges and future directions of Verifiable Delay Function and Its Blockchain-Related Application should be explained in detail as a separate section.

Author response: Thank you for your careful inspection. Regarding the suggestion about the context, we have followed your suggestions and supplied the open challenges and future directions of Verifiable Delay Function and Its Blockchain-Related Application in Section 6. The details are as follows:

Verifiable delay functions are widely used in blockchains. However, verifiable delay functions based on finite groups of unknown order are insecure against an adversary with access to a quantum computer. Quantum computers can easily compute the order of a group using Shor’s algorithm, making it easy to break the application of verifiable delay functions in blockchains. In addition, the post-quantum secure verifiable delay function based on super-singular elliptic curves needs to be verified by SNARG, and the structure is in its infancy and can only be limited to the field of elliptic curves. Therefore, it is an open problem to research a verifiable delay function that has a simple structure in quantum computers and can be safely applied to blockchain.

Round 2

Reviewer 2 Report

The authors have addressed all of my queries. I believe the paper can be accepted for publication.